# Effects of UVC Treatment on Biofilms of *Escherichia coli* Strains Formed at Different Temperatures and Maturation Periods

**DOI:** 10.3390/foods14173091

**Published:** 2025-09-03

**Authors:** Myounghyeon Kyoung, Jae-Ik Lee, Sang-Soon Kim

**Affiliations:** 1Department of Food Engineering, Dankook University, Cheonan 31116, Republic of Korea; 2Department of Food and Nutrition, College of Biomedical and Health Science, Konkuk University, Chungju 27478, Republic of Korea; jaeiklee@kku.ac.kr; 3School of Animal & Food Sciences and Marketing, Konkuk University, Seoul 05029, Republic of Korea

**Keywords:** metagenome analysis, *Escherichia coli*, biofilm, UVC, API test, environmental stress

## Abstract

In the present study, the biofilm formation and ultraviolet-C (UVC) resistance characteristics of *Escherichia coli* isolated from an occluded biliary stent were compared with those of four *E. coli* O157:H7 strains (ATCC 35150, 43889, 43890, and 43895). To evaluate biofilm formation, the *E. coli* isolated from a stent and four *E. coli* O157:H7 strains were incubated at 37, 25, and 15 °C for 7 days, revealing that peak biofilm formation occurred at 37 °C (day 1), 25 °C (day 3), and 15 °C (day 5), with the stent-isolated strain consistently exhibiting significantly higher biofilm cell counts than the others (*p* < 0.05). The UVC treatment was less effective at reducing viable biofilm cells as the formation temperature decreased, with the stent-isolated *E. coli* biofilm formed at 15 °C showing the lowest reduction levels. Exopolysaccharide quantification revealed that all *E. coli* strains produced more extracellular polymeric substances (EPSs) at lower temperatures, with the stent-isolated *E. coli* biofilm formed at 15 °C showing significantly higher EPS levels than the other strains (*p* < 0.05), potentially explaining its greater UVC resistance. Based on these results, it was confirmed that the biofilm formed by the *E. coli* isolated from the stent at 15 °C exhibited the highest resistance to UVC, which can be attributed to its elevated exopolysaccharide production. This study demonstrates that both temperature and maturation period significantly influence *E. coli* biofilm characteristics and provides valuable insights into *E. coli* isolated from the stent, which may pose a risk of cross-contamination in food-related environments.

## 1. Introduction

*Escherichia coli* is a Gram-negative, non-spore-forming, facultatively anaerobic, rod-shaped bacterium belonging to the Enterobacteriaceae family. First described in 1885, most *E. coli* strains are considered part of the normal intestinal microbiota of humans and animals and do not pose health concerns [1]. However, certain *E. coli* strains possess virulence factors that can cause a range of diseases, including urinary tract infections, intestinal and diarrheal diseases, sepsis, and meningitis [2,3]. Pathogenic *E. coli* can be classified based on serogroup, pathogenic mechanisms, clinical symptoms, and virulence factors [4]. Among these, enterohemorrhagic *Escherichia coli* O157:H7, first recognized in 1982 as an intestinal pathogen and a significant cause of foodborne illness, produces Shiga-like toxins (*Stxs*) and can cause hemorrhagic colitis (HC), as well as the life-threatening hemolytic uremic syndrome (HUS) [5]. *E. coli* O157:H7 is characterized by a low infectious dose, with infection being possible following the ingestion of fewer than 50 colony-forming units (CFU) [6], and outbreaks have been linked to undercooked meat products, raw milk, contaminated produce, and contaminated drinking water [7]. According to data from the Ministry of Food and Drug Safety of Korea, pathogenic *E. coli* was the leading cause of bacterial foodborne illness from 2015 to 2021 [8]. In 2023, there were 46 reported outbreaks and 2287 patients, accounting for 23.7% of all foodborne illness cases in South Korea [9]. In the United States, the Centers for Disease Control and Prevention (CDC) reported 3351 cases of pathogenic *E. coli* infection in 2023 [10]. Additionally, in 2024, an outbreak of *E. coli* O157:H7 linked to contaminated hamburgers resulted in at least 104 infections across 14 states in the United States [11].

Several reports indicated that various strains of *E*. *coli* can form biofilms on food-contact surfaces and a wide range of medical devices, which can lead to cross-contamination and subsequent infection by pathogenic bacteria during food processing [12,13]. Furthermore, according to the National Institutes of Health (NIH), it has been estimated that 80% of all human infections are associated with biofilms [14]. A biofilm is a microbial community formed by bacteria embedded within an extracellular polymeric substance (EPS) composed of self-secreted polysaccharides, proteins, and extracellular DNA (eDNA) [15]. The formation of *E. coli* biofilms occurs through several stages, including initial attachment, early development, maturation, and dispersion [16,17]. Compared with planktonic cells, the biofilm matrix provides structural stability and enhances resistance to environmental stressors such as disinfectants, antibiotics, temperature fluctuations, and ultraviolet radiation [18]. Numerous studies have demonstrated that *E. coli* strains can grow under various environmental conditions and on a wide range of surface types, including both abiotic and biotic surfaces [19,20]. Biofilms formed on food-contact surfaces are associated with bacterial cross-contamination of food products [21], and *E. coli* species capable of biofilm formation can serve as sources of such cross-contamination [22]. Furthermore, it has been reported that *E. coli* can form biofilms on various medical devices, such as urinary and intravascular catheters, prosthetic joints, and stents, which are major contributors to increased morbidity and mortality [22].

Food scientists have attempted to detect the foodborne pathogens rapidly and precisely [23,24] and to inactivate these pathogens effectively [25]. Various technologies, including heat treatment, irradiation, microwave, pulsed electric fields, plasma [26] magnetic fields, chlorine dioxide [27] and high pressure, have been evaluated for the inactivation of pathogenic microorganisms [28]. Among these methods, ultraviolet (UV) radiation has been shown to be a simple and effective method for microbial destruction [29] and is commonly used to achieve microbiological pasteurization to safe levels without altering color, flavor, or pH [30]. The lethal effect of UV radiation results from photolytic reactions that cause structural modifications in nucleic acids (DNA and RNA), thereby preventing transcription and replication. This leads to physiological disruptions that induce cell death [31]. In recent years, UV equipment has become widely adopted and is commonly used to control microorganisms in various settings, and metagenomic analysis can be used effectively to identify the microbiota differences before and after UV-C treatment [32].

The primary function of the bile duct is to transport and release bile, a digestive fluid produced and secreted by the liver [33]. In addition to facilitating the elimination of lipid-soluble waste products, bile aids in the digestion and absorption of fats and fat-soluble vitamins [34]. Biliary stricture is a major cause of impaired bile flow [35], which can lead to serious complications such as hepatic dysfunction, renal failure, malnutrition, and infection [36]. Biliary strictures can result from benign causes, such as inflammation, or from malignant causes, such as cancer. Endoscopic retrograde cholangiopancreatography (ERCP) with biliary stent placement is widely utilized to treat these strictures [37]. However, even after stent insertion to maintain bile flow, stent occlusion may occur due to the accumulation of biliary sludge and the formation of microbial biofilms within the stent. Therefore, it is recommended that biliary stents be replaced with new ones every 3 to 6 months [38]. In this study, the biofilm formation characteristics of microorganism isolated from the biliary stents were investigated and compared with the pathogenic *E. coli* strains.

In this study, (1) metagenome analysis was employed to identify the microbial communities forming biofilms on biliary stents obtained from patients, followed by isolation and identification of *Escherichia coli* using the Analytical Profile Index (API) test. (2) The biofilm formation abilities of the *E. coli* isolated from a stent and four *E. coli* O157:H7 strains were compared by inoculating each strain onto polyethylene (PE) coupons and incubating them at 37 °C, 25 °C, and 15 °C for up to 7 days to assess the effects of maturation temperature and duration. (3) The efficacy of UVC treatment against mature *E. coli* biofilms formed at different temperatures was evaluated for each strain. (4) Exopolysaccharide quantification of the extracellular polymeric substances (EPSs) in the biofilms was conducted to elucidate strain-specific differences in biofilm resistance under various growth conditions to identify the reason for different UVC resistances. We aimed to investigate the biofilm formation and UVC resistance of *Escherichia coli* isolated from occluded biliary stents, in comparison with *E. coli* O157:H7 reference strains, under various temperature and maturation conditions. Using metagenome analysis and API profiling, we identified the dominant strain, evaluated its biofilm development on polyethylene surfaces, and quantified exopolysaccharide levels to elucidate the mechanism of UVC resistance.

## 2. Materials and Methods

### 2.1. Metagenome Analysis of Biliary Stents

#### 2.1.1. DNA Extraction and 16S rRNA Gene Amplification

Bile samples and biliary stents from patients were collected by Dankook University Hospital, Cheonan, Republic of Korea with ethical approval obtained from the Faculty of Dankook University Hospital Review Board (IRB No: 2023-06-004). The outside of the stent was washed with 70% ethanol. The washed stent was cut into uniform pieces (approximately 0.5 cm) using a medical scalpel, and vortexed with glass beads. DNA samples were extracted using the Qiagen Fast DNA Stool Mini Kit (Qiagen, Hilden, Germany). After washing the exterior of the stent, the stent was cut into small pieces, and the microorganisms from the inner surface were collected for DNA extraction. The concentration and purity of the extracted DNA were assessed using a Qubit 4.0 (Invitrogen Corporation, Carlsbad, CA, USA) and a Nanodrop™ Fluorospectrometer (Thermo Fisher Scientific Inc., Waltham, MA, USA) according to the manufacturers’ instructions. Among the 4 samples, a sample with the highest quality was further analyzed for the 16S rRNA metagenome sequencing. The bacterial 16S rRNA V3–V4 hypervariable regions were amplified via polymerase chain reaction (PCR) using KAPA HIFI Hot Start Ready Mix (2×) (Roche Cat. No. 07958935001; Basel, Switzerland). The following primers, containing Illumina overhang adapter sequences, were used for amplification of the 16S rRNA gene:(341F: 5′-TCGTCGGCAGCGTCAGATGTGTATAAGAGACAGCCTACGGGNGGCWGCAG-3′)(806R: 5′-GTCTCGTGGGCTCGGAGATGTGTATAAGAGACAGGACTACHVGGGTATCTAATCC-3′)

Sequencing was performed using the Illumina NextSeq system (Illumina, San Diego, CA, USA) according to the manufacturer’s instructions. The amplified products (amplicons) were sequenced on the NextSeq 2000 platform with 300 bp paired-end reads, targeting a sequencing depth of over 100,000 reads per sample.

#### 2.1.2. Sequencing Data Processing and Bioinformatics Analysis

Sequencing data generated from the Illumina NextSeq system were processed using the QIIME2 software package version 2023.2. Trimming, quality filtering, paired-end merging, and denoising were performed with the DADA2 plugin (dada2 denoise-paired option) [39]. Amplicon sequence variants (ASVs) were defined as the final output of DADA2 and were used for subsequent analyses. Taxonomic classification of ASVs was conducted using the Naïve Bayes classifier in QIIME2 with reference to the SILVA database v138. ASVs were aligned using the multiple sequence alignment tool MAFFT 40], and a phylogenetic tree was constructed with the FastTree plugin [40].

### 2.2. API Test Method

In this study, the API 20E kit was used to isolate and identify microorganisms forming biofilms on biliary stents obtained from patients. The API 20E kit, suspension medium, additional reagents, and mineral oil were purchased from BioMerieux (Marcy-L’Etoile, France). All components of the kit, except for mineral oil, were stored at 5 °C until use. To isolate microorganisms from the patients’ biliary stents, each stent was first incubated in Luria–Bertani (LB) broth (Difco, Franklin Lakes, NJ, USA) at 37 °C for 6 h. The resulting culture was then streaked onto MacConkey agar (Oxoid Ltd., Basingstoke, Hampshire, UK) and incubated at 37 °C for 24 h. From the colonies observed on MacConkey agar, three pink colonies and three colorless colonies were selected and identified using the API 20E kit, which is commonly employed for the identification of enteric and other Gram-negative rods [41]. A single representative colony was subsequently inoculated into 5 mL of tryptic soy broth (TSB) and incubated at 37 °C for 24 h before being subjected to identification using the API 20E kit. For the API 20E kit, the cupules were inoculated with the bacterial suspension according to the manufacturer’s instructions. The kit was then incubated at 37 °C for 24 h. After incubation, additional API 20E reagents were added according to the manufacturer’s instructions. The results were analyzed using the API reading chart provided on the API website, with the sequence of positive (+) and negative (−) reactions for each strain entered accordingly. Identification and validation of the strains were performed using the API software, and the results were expressed as % identification (ID).

### 2.3. Bacterial Cell Preparation

The *Escherichia coli* used in this experiment was isolated from a biliary stent and identified as *E. coli* through the API test. Additionally, four strains each of *E. coli* O157:H7 (ATCC 35150, ATCC 43889, ATCC 43890, and ATCC 43895) obtained from the bacterial culture collection of the School of Food Engineering, Dankook University (Cheonan, Republic of Korea) were used in the present study. Each strain of *E. coli* was pre-cultured in 5 mL of tryptic soy broth (TSB, Difco, Franklin Lakes, NJ, USA) at 37 °C for 24 h. After incubation, the cultures were mixed using a vortex mixer (Vortex VM-10, DAIHAN Scientific Company, Seoul, Republic of Korea) and centrifuged (Union 55R, Hanil science Company, Gimpo, Republic of Korea) at 4470× *g* for 20 min at 4 °C [42]. The final pellets were resuspended in sterile phosphate-buffered saline (PBS, pH 7.4) corresponding to approximately 7–8 log CFU/mL.

### 2.4. Coupon Preparation

Polyethylene (PE) coupons (2 × 2 × 1 cm, KUMJOUNG EPS, Ansan, Republic of Korea) were used in the experiment. The coupons were disinfected by immersing them in 70% ethanol for 60 min, then rinsed with sterile distilled water. They were further sterilized by autoclaving at 121 °C for 15 min. After sterilization, the coupons were dried and stored in a dry oven at 55 °C.

### 2.5. Biofilm Formation Assay

Each prepared PE coupon was immersed in a sterile 50 mL conical centrifuge tube (SPL Lifesciences, Pocheon, Republic of Korea) containing 30 mL suspensions of *E. coli* in PBS [43]. The coupons with bacterial cell suspensions were incubated at 4 °C for 24 h to facilitate cell attachment [44]. The coupons were then immersed in 30 mL of TSB in 50 mL conical centrifuge tubes and incubated at 15 °C, 25 °C, and 37 °C for 1 to 7 days.

### 2.6. Extracellular Polymeric Substance (EPS) Component Assay

#### 2.6.1. Extraction of EPS in *E. coli* Biofilms

The biofilm-matured PE coupons at each temperature were washed with sterile distilled water using sterile forceps to remove loosely attached cells. The biofilms were then scraped off using a sterile cell scraper (Jet Biofil, Guangzhou, China) and 1 mL of 25 mM EDTA solution. The scraped biofilm cell suspensions were collected in a sterile 1.5 mL microcentrifuge tube (Axygen, Union City, CA, USA) and incubated at 4 °C for 1.5 h. After incubation, the suspensions were centrifuged at 12,000× *g* for 15 min, and the supernatants were collected and filtered with a 0.22 μm membrane filter (Hyundaimicro, Seoul, Republic of Korea). The extracted EPSs were stored at −80 °C prior to carbohydrate analysis.

#### 2.6.2. Quantification of Carbohydrate Concentrations in Extracted EPSs

The total carbohydrate content of the extracted EPS compounds was measured using the acid hydrolysis method with sulfuric acid. A mixture of 30 μL of extracted EPSs and 150 μL of 98% sulfuric acid was heated at 90 °C for 15 min in water bath. After cooling the mixture at 4 °C for 3 min, it was prepared in a 96-well plate, and the absorbance was measured at 490 nm using a spectrophotometer (Spectramax ABS Plus, Molecular Devices, San Jose, CA, USA). Glucose was used as the standard to quantify the total carbohydrate content.

### 2.7. UVC Treatment

In this study, the UVC device described by [45] was used. A germicidal ultraviolet lamp (GL20T5, Sankyo Denki Co., Kanagawa, Japan) with a peak emission wavelength of 253.7 nm was used to inactivate *E. coli* biofilm cells matured at 15 °C, 25 °C, and 37 °C. The distance between the coupons and the UVC lamp was set at 17 cm, and the irradiation intensity (1.58 mW/cm^2^) was measured using a UVC meter (HPL220UVC-254, HOPOOCOLOR, Hangzhou, China). The biofilm-matured PE coupons at each temperature were washed with sterile distilled water using sterile forceps to remove loosely attached cells. To ensure selective exposure, the non-exposed surfaces of the coupons were treated with 70% ethanol. Coupons were irradiated with UVC at 30 s intervals from 0 to 180 s. Immediately after UVC treatment, the coupons were transferred to 50 mL conical centrifuge tubes containing 20 mL of sterile PBS and 1 g of glass beads and vortexed for 1 min using a vortex mixer. For enumeration, cell suspensions were ten-fold serially diluted in 9 mL of PW. After that, 100 µL aliquots were spread-plated onto Sorbitol MacConkey agar (Difco), followed by incubation for 24 h at 37 °C. After incubation, typical colonies (pink colonies for isolated *E. coli* and white colonies for *E. coli* O157:H7) were counted.

### 2.8. Statistical Analysis

Biofilm quantification, UVC treatment, and EPS quantification experiments were performed in triplicate, and the results are presented as the mean ± standard deviation. Figures were generated using the Prism 8.0 software (GraphPad Software Inc., La Jolla, CA, USA). Statistical significance was assessed using analysis of variance (ANOVA) and Duncan’s multiple range test with the IBM SPSS 26 software package (SPSS Inc., Chicago, IL, USA). Differences were considered statistically significant at a 95.0% confidence level (*p* < 0.05). Schematic diagram of experimental design was shown in Figure 1.

## 3. Results and Discussion

### 3.1. Metagenome Analysis of Biliary Stents and Identification of Microorganisms Using the API Test

Biliary stents maintain ductal patency in patients with biliary strictures. However, stent occlusion can frequently occur due to the formation of biofilms by microorganisms. Stent occlusion is associated with the development of jaundice and bacterial cholangitis, often accompanied by polymicrobial infections. Therefore, information on the microbial communities forming biofilms within biliary stents is essential for the appropriate use of antibiotics and the implementation of effective treatment strategies. To identify the causative microbial communities responsible for biofilm formation and stent occlusion in patients, 16S rRNA amplicon sequencing was performed on the biliary stents obtained from patients. The results of the genus-level taxonomic analysis of the patients’ stents are presented in Figure 2. Enterobacteriaceae accounted for the highest proportion at 47%, with the genus *Escherichia* and *Shigella* representing 9% of the total. *Enterococcus*, belonging to the order Lactobacillales, comprised 25%. *Veillonella*, commonly found in the oral mucosa and intestine, accounted for 9%, while *Selenomonas*, typically residing in the gastrointestinal tract, represented 7%. The microbial communities primarily involved in the process of stent occlusion vary across previously reported studies. For example, in a study by the Cacai et al. [46], metagenome sequencing was used to analyze the microbial communities present in the stents of 22 patients, and *Escherichia coli*, *Klebsiella pneumoniae*, *Enterococcus faecalis*, and *Lactobacillus* spp. were identified as the predominant species. The other study by Vaishnavi et al. [47] identified the microbial communities in stents from 81 patients were analyzed using polymerase chain reaction (PCR) and denaturing gradient gel electrophoresis (DGGE), with *Citrobacter*, *Klebsiella*, *Staphylococcus*, *Serratia*, and *Escherichia coli* being identified as the predominant genera. Similarly Blanco-Míguez et al. [48] employed shotgun metagenome sequencing to examine the microbial communities in stents from 56 patients, reporting *Streptococcus anginosus*, *Escherichia coli*, and *Enterococcus faecalis* as the dominant species.

Based on the results of this experiment and previous studies, Enterobacteriaceae were identified as the predominant microbial group responsible for stent occlusion. Therefore, to isolate microorganisms forming biofilms on stents, MacConkey agar—a selective and differential medium that promotes the growth of Gram-negative enteric bacteria and differentiates lactose fermenters—was used. From the colonies observed on MacConkey agar, three pink colonies and three colorless colonies were selected and identified using the API 20E kit, which is commonly employed for the identification of enteric and other Gram-negative rods. The pink colonies were identified as *Escherichia coli* (99.9%), *Klebsiella pneumoniae* (97.3%), and *Enterobacter cloacae* (98.6%), while the colorless colonies were identified as *Aeromonas hydrophila/caviae/sobria* (94.9%) and *Serratia ficaria* (97.1%) (Appendix A). In this study, metagenome analysis was utilized to identify the major microbial communities responsible for biliary stent occlusion, and microorganisms forming biofilms on the stent were isolated and identified using the API test. The results revealed that Enterobacteriaceae were the predominant group among the microbial communities causing stent occlusion. Furthermore, the microorganisms isolated and identified from the stent—*Escherichia coli*, *Klebsiella pneumoniae*, and *Enterobacter cloacae*—all belonged to the family Enterobacteriaceae, which is consistent with findings from previous studies. Therefore, a subsequent experiments were conducted to evaluate the biofilm formation abilities and the effects of UVC treatment among *E. coli* strains, including the isolated *Escherichia coli* and four *E. coli* O157:H7 strains, under different maturation temperatures and durations.

### 3.2. Biofilm Formation at Different Temperatures and Maturation Periods

To assess the biofilm formation abilities of the *E. coli* isolated from stent and *E. coli* O157:H7 ATCC 35150, 43889, 43890, and 43895 under different maturation temperatures and durations, experimental conditions were set at 15 °C, 25 °C, and 37 °C, with maturation periods of 1, 3, 5, and 7 days. These temperature ranges were selected based on their relevance to food processing environments (15 °C), room temperature (25 °C), and optimal microbial growth conditions for *E. coli* (37 °C). The initial cell counts of biofilms formed by the *E. coli* isolated from the stent and *E. coli* O157:H7 ATCC 35150, 43889, 43890, and 43895 were 4.57 ± 0.17, 4.42 ± 0.14, 4.67 ± 0.19, 4.73 ± 0.07, and 4.35 ± 0.11 log CFU/cm^2^, respectively. The biofilm formation abilities of each *E. coli* strain according to maturation temperature and duration are presented in Table 1. In case of the *E. coli* isolated from the stent, the highest biofilm cell counts were observed as 6.89 ± 0.23 log CFU/cm^2^ after 1 day at 37 °C, 7.14 ± 0.19 log CFU/cm^2^ after 3 days at 25 °C, and 7.06 ± 0.13 log CFU/cm^2^ after 5 days at 15 °C, respectively. There were no significant differences among the highest biofilm cell counts at each temperature. At 37 °C, the biofilm cell counts of the *E. coli* isolated from the stent peaked at 1 day and gradually decreased, reaching the lowest count at day 7 (*p* < 0.05). Specifically, the cell count was 6.89 ± 0.23 log CFU/cm^2^ at day 1 and significantly decreased to 6.02 ± 0.40 log CFU/cm^2^ at day 7. For *E. coli* O157:H7 ATCC 35150, 43889, 43890, and 43895, the highest biofilm cell counts at 37 °C were observed after 1 day of maturation, with values of 6.04 ± 0.24, 6.22 ± 0.29, 6.13 ± 0.25, and 6.36 ± 0.40 log CFU/cm^2^, respectively. At 25 °C, the highest counts were recorded after 3 days of maturation, showing 6.09 ± 0.18, 6.53 ± 0.10, 6.29 ± 0.28, and 6.19 ± 0.34 log CFU/cm^2^, respectively. At 15 °C, the peak biofilm cell counts were 6.26 ± 0.17, 6.29 ± 0.13, 6.16 ± 0.08, and 6.30 ± 0.38 log CFU/cm^2^, respectively, for ATCC 35150, 43889, 43890, and 43895. There were no significant differences in the maximum biofilm cell counts among the different temperatures. Similarly to the *E. coli* isolated from the stent, the biofilm cell counts of *E. coli* O157:H7 strains at 37 °C peaked at day 1 and gradually decreased during the maturation period, although these changes were not statistically significant. Specifically, the cell counts at 1 day were 6.04 ± 0.24, 6.22 ± 0.29, 6.13 ± 0.25, and 6.36 ± 0.40 log CFU/cm^2^, which decreased to 5.65 ± 0.29, 5.71 ± 0.55, 5.77 ± 0.41, and 6.07 ± 0.65 log CFU/cm^2^ at 7 days, respectively.

Overall, the biofilm cell counts of all *E. coli* strains increased at 15 °C, while relatively stable counts were observed at 25 °C. Based on the most mature biofilm cell counts at each temperature, the *E. coli* isolated from the stent, as well as *E. coli* O157:H7 ATCC 43889 and 43890, exhibited the highest biofilm-forming abilities at 25 °C. Similar biofilm formation characteristics were observed in the study by Ma & Bumunang [49], who reported that certain *E. coli* strains maintained stable cell counts at 22 °C during the maturation period, while cell counts increased at 13 °C. The other study also demonstrated that *E. coli* forms more biofilm at 25 °C and 37 °C compared with 12 °C [50]. These trends have also been reported by Nesse et al. [51] and Marti et al. [52], who found that *E. coli* isolates exhibited the highest biofilm formation ability at 28 °C among the tested temperatures of 12 °C, 28 °C, and 37 °C.

In contrast, for all *E. coli* strains, the highest biofilm cell counts at 37 °C were observed after 1 day, followed by a gradual decrease to the lowest counts on day 7. This phenomenon may be attributed to the fact that *E. coli* exhibits optimal growth at 37 °C; abundant nutrients and optimal temperature at the early stage of incubation lead to increased metabolic rates and nutrient consumption. Consequently, nutrient depletion and the accumulation of metabolic waste products occur more rapidly at 37 °C than at lower temperatures [53]. Similar results showing a decrease in *E. coli* biofilm cell counts with maturation time at 37 °C have also been reported in [54,55].

Collectively, all five strains exhibited strong biofilm formation abilities, with the most mature biofilm cell count observed at maturation at 37 °C for 1 day, at 25 °C for 3 days, and at 15 °C for 5 days. Except for the biofilm of *E. coli* ATCC 43895 matured at 37 °C, the *E. coli* isolated from the stent consistently showed significantly higher mature biofilm cell counts than the other *E. coli* strains at all tested temperatures (Figure 3). These results suggest that the *E. coli* isolated from the stent possesses a greater capacity to form robust biofilms under various environmental conditions of temperature and maturation periods compared with the other *E. coli* strains.

### 3.3. Resistance of E. coli Biofilm to UVC Treatment

This experiment was conducted to compare the effects of biofilm maturation temperature on the UVC resistance of biofilms formed by the *E. coli* isolated from the stent and *E. coli* O157:H7 ATCC 35150, 43889, 43890, and 43895. Based on the previous biofilm formation results, all five *E. coli* strains produced the most mature biofilms after maturation at 37 °C for 1 day, at 25 °C for 3 days, and at 15 °C for 5 days. Therefore, UVC treatment was applied to the mature biofilms of each strain at 1 day (37 °C), 3 days (25 °C), and 5 days (15 °C) in 30 s intervals for up to 180 s.

UVC treatment reduced the biofilm cell counts of each *E. coli* strain as the exposure time increased (Table 2). For some *E. coli* strains, biofilms formed at 37 °C or 25 °C exhibited greater reductions in cell counts compared with those formed at 15 °C. For example, after 180 s of UVC exposure, the reduction levels of the *E. coli* isolated from the stent were 3.46 ± 0.29 log CFU/cm^2^ at 37 °C, 2.24 ± 0.05 log CFU/cm^2^ at 25 °C, and 1.85 ± 0.41 log CFU/cm^2^ at 15 °C. Additionally, the reduction levels of *E. coli* O157:H7 ATCC 35150 biofilms were 3.45 ± 0.33 log CFU/cm^2^, 2.95 ± 0.38 log CFU/cm^2^, and 2.34 ± 0.27 log CFU/cm^2^, with maturation temperatures of 37, 25, and 15 °C, respectively, after 180 s of UVC treatment. In particular, the reduction in biofilm cell counts for the *E. coli* isolated from the stent was significantly lower in biofilms matured at 25 °C and 15 °C compared with those matured at 37 °C (*p* < 0.05). Similarly, for *E. coli* O157:H7 ATCC 35150, the reduction levels were significantly lower in biofilms matured at 15 °C than at 37 °C (*p* < 0.05). However, for *E. coli* O157:H7 ATCC 43889, 43,890, and 43895, the biofilm maturation temperature showed no significant effects on UVC resistance after 180 s of UVC treatment (*p* > 0.05).

Previous study [56] compared the survival of *E. coli* O157:H7 ATCC 43895 after forming biofilms on lettuce surfaces and subjected them to LDPE for 24 and 48 h, followed by pulsed UV treatment, and they reported that biofilms matured for 48 h exhibited greater resistance to pulsed UV than those matured for 24 h. Previous study [57] analyzed the characteristics and resistance of *E. coli* O157:H7 biofilms formed at 12 °C and 22 °C, and they demonstrated that biofilms formed at 12 °C contained higher levels of extracellular polymeric substances (EPSs) compared with those formed at 22 °C, suggesting that biofilms matured in low-temperature environments may possess greater resistance. Previously, Lee et al. [58] also reported that *E. coli* produces higher levels of indole signaling molecules in low-temperature environments, which leads to increased EPS production and greater structural density of biofilms, thereby enhancing resistance to external stresses such as heat and disinfectants.

Biofilms of the *E. coli* isolated from the stent, as well as *E. coli* O157:H7 ATCC 35150, matured at 15 °C exhibited smaller reductions following UVC treatment compared with those matured at other temperatures, indicating increased UVC resistance at 15 °C. In particular, the *E. coli* isolated from the stent and matured at 15 °C showed significantly less reduction in response to UVC treatment than the other strains, with a particularly significant difference compared with *E. coli* ATCC 43890 and 43895 (*p* < 0.05).

These results indicated that maturation temperature influences UVC resistance, with biofilms matured at lower temperatures exhibiting greater resistance to UVC compared with those matured at higher temperatures. Furthermore, these findings suggest that *E. coli* isolated from stent biofilms formed at low temperatures possess greater resistance than other *E. coli* strains under the same conditions.

### 3.4. Quantification of EPSs in E. coli Biofilms at Different Temperatures

EPSs account for 50–90% of the total organic matter in biofilms and are primarily composed of various high-molecular-weight substances, including polysaccharides, proteins, and eDNA [59]. Furthermore, EPSs can facilitate the cohesion of bacterial cells and provide protection against external environmental stresses, which served as the rationale for this experiment [60]. To investigate the UVC resistance of biofilms formed at 15 °C, quantification of exopolysaccharides within the EPSs was performed.

The quantification results of exopolysaccharides in the biofilms of the *E. coli* isolated from the stent and *E. coli* O157:H7 ATCC 35150, 43889, 43890, and 43,895 are presented in Table 3. The exopolysaccharide content increased as the temperature decreased in the biofilms of all *E. coli* strains. For example, in the most mature biofilms at each temperature, the exopolysaccharide contents in the *E. coli* isolated from the stent and *E. coli* O157:H7 ATCC 35150, 43889, 43890, and 43895 were 1.33 ± 0.09, 1.14 ± 0.10, 1.19 ± 0.12, 1.17 ± 0.12, and 1.16 ± 0.09 μg/cm^2^ at 37 °C, respectively. At 25 °C, the values were 1.74 ± 0.03, 1.54 ± 0.03, 1.58 ± 0.03, 1.51 ± 0.02, and 1.54 ± 0.02 μg/cm^2^, respectively. At 15 °C, the exopolysaccharide contents were 2.58 ± 0.03, 1.68 ± 0.01, 1.70 ± 0.02, 1.63 ± 0.01, and 1.73 ± 0.02 μg/cm^2^, respectively. Although the exopolysaccharide contents of the *E. coli* O157:H7 ATCC 35150, 43889, 43890, and 43,895 biofilms increased as the temperature decreased, these differences were not statistically significant, nor were there significant differences according to maturation period. However, the exopolysaccharide content of the biofilm of the *E. coli* isolated from the stent increased significantly as the temperature decreased (*p* < 0.05). Furthermore, at 15 °C, the biofilm of the *E. coli* isolated from the stent exhibited significantly higher exopolysaccharide content than the biofilms of all other *E. coli* strains at all maturation periods (*p* < 0.05).

Exopolysaccharides constitute a larger proportion of the biofilm matrix than proteins or eDNA [61]. Li et al. [62] also reported that the main EPS components of *E. coli* are polysaccharides such as colanic acid, poly-β-1,6-N-acetyl-D-glucosamine (PGA), and cellulose and that more than 78% of the EPSs produced by *E. coli* consist of exopolysaccharides. Vogeleer et al. [63] reported that STEC is capable of forming biofilms even at low temperatures ranging from 4 to 15 °C, and that EPSs, particularly colanic acid, enhance resistance to osmotic and oxidative stress. Ryu & Beuchat [64] investigated the effects of EPSs and curli production on the resistance of *E. coli* O157:H7 biofilms to the disinfectant chlorine (Cl_2_) and found that at low temperatures (12 °C), EPSs, and curli production significantly increased the resistance of *E. coli* O157:H7 biofilms to the disinfectantTherefore, these results collectively indicate that the increased resistance to UVC treatment observed at 15 °C can be attributed to the elevated production of exopolysaccharides. In particular, the *E. coli* isolated from the stent produced the highest level of exopolysaccharides under 15 °C conditions compared with the other strains, suggesting that this is associated with its greater resistance. Recently, Lee et al. [25] reported higher susceptibility of *E. coli* O157:H7 biofilm to the amalgam lamp compared to the conventional LP lamp. Therefore, amalgam lamps would be an effective alternative to the conventional LP lamps to control *E. coli* O157:H7 biofilms.

## 4. Conclusions

In this study, *E*. *coli* isolated from an occluded biliary stent was compared with four reference *E. coli* O157:H7 strains to assess differences in biofilm formation, UVC resistance, and exopolysaccharide production under varying maturation temperatures. The stent-isolated strain consistently formed more robust biofilms and exhibited significantly higher resistance to UVC, particularly at 15 °C, where it also produced the highest levels of exopolysaccharides. These findings suggest that low-temperature environments enhance biofilm resistance through increased EPS production. The results highlight the potential public health risks posed by biofilm-forming *E. coli* strains isolated from clinical settings, especially under suboptimal environmental conditions. This study provides new insights into the stress tolerance mechanisms of *E. coli* biofilms and underscores the importance of tailored disinfection strategies for biofilms formed at low temperatures.

## Figures and Tables

**Figure 1 foods-14-03091-f001:**
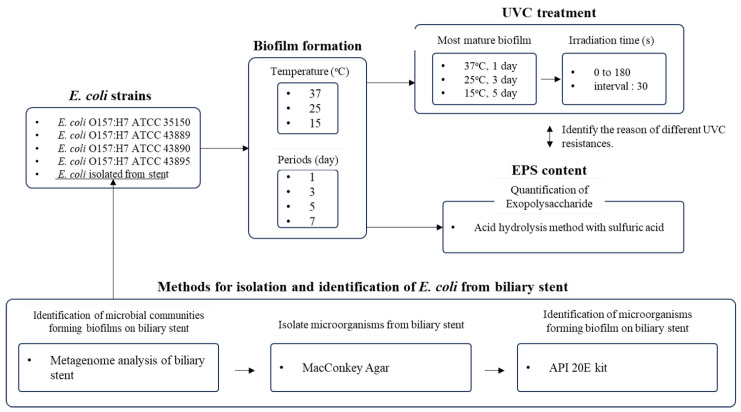
Schematic diagram of experimental design including methods for isolation an identification of *E. coli* from biliary stent, biofilm formation strategy, UVC treatment condition, and quantification of exopolysaccharide.

**Figure 2 foods-14-03091-f002:**
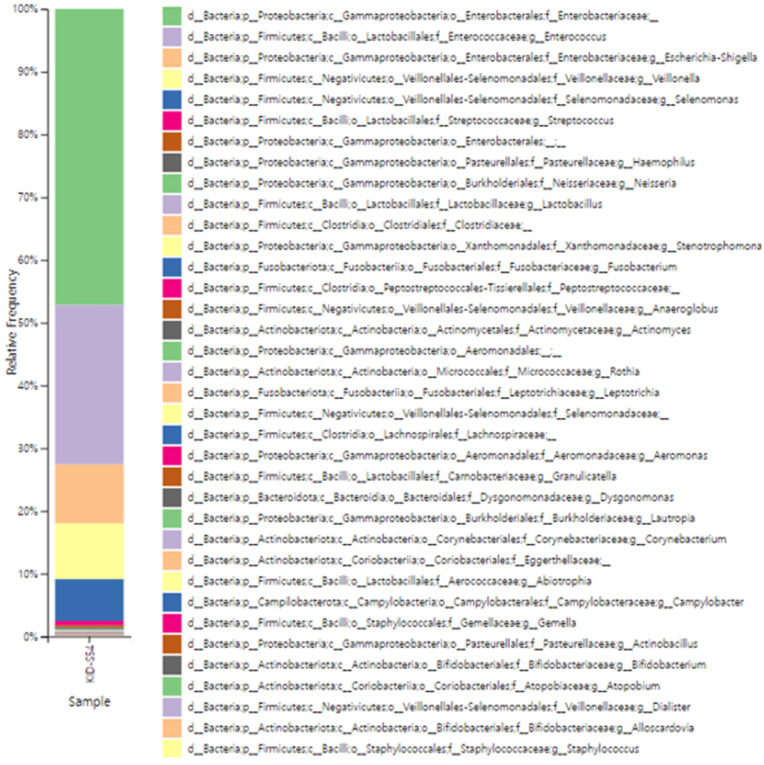
Taxonomic analysis of biliary stents using 16S rRNA amplicon sequencing.

**Figure 3 foods-14-03091-f003:**
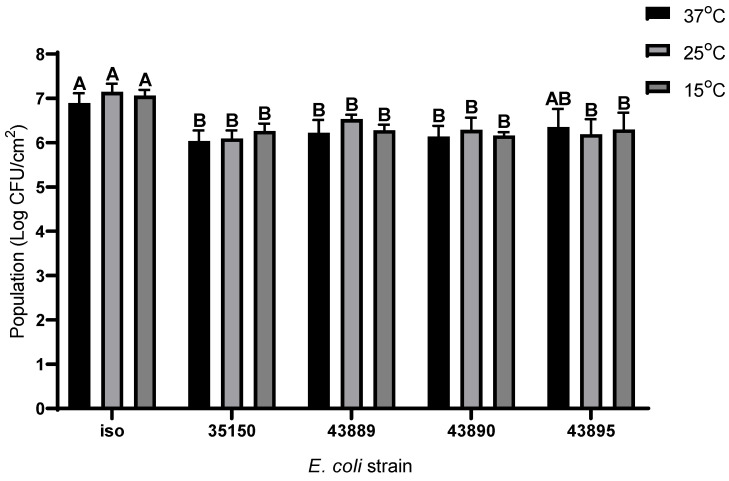
The highest biofilm population levels (log CFU/cm^2^) of five *E. coli* matured on polyethylene at 37 °C, 25 °C, and 15 °C. Different letters represent significant differences between *E. coli* strain within the same growth temperature.

**Table 1 foods-14-03091-t001:** Population (log CFU/cm^2^) of *E. coli* isolated from a biliary stent and *E. coli* O157:H7 (ATCC 35150, ATCC 43889, ATCC 43890, and ATCC 43895) biofilm on polyethylene at 37 °C, 25 °C, and 15 °C for 1, 3, 5, and 7 days ^1,2^.

Biofilm Maturation Temperature	*E. coli* Strain	Biofilm Maturation Time			
		1 Day	3 Day	5 Day	7 Day
**37 °C**	iso	6.89 ± 0.23 ^Aa^	6.54 ± 0.23 ^Aab^	6.18 ± 0.64 ^Ab^	6.02 ± 0.40 ^Ab^
	35150	6.04 ± 0.24 ^Ba^	5.81 ± 0.13 ^Aa^	5.73 ± 0.40 ^Aa^	5.65 ± 0.29 ^Aa^
	43889	6.22 ± 0.29 ^ABa^	5.86 ± 0.48 ^Aa^	5.64 ± 0.26 ^Aa^	5.71 ± 0.55 ^Aa^
	43890	6.13 ± 0.25 ^Ba^	5.91 ± 0.26 ^Aa^	5.93 ± 0.16 ^Aa^	5.77 ± 0.41 ^Aa^
	43895	6.36 ± 0.40 ^ABa^	6.25 ± 0.36 ^Aa^	6.11 ± 0.61 ^Aa^	6.07 ± 0.65 ^Aa^
**25 °C**	iso	6.54 ± 0.47 ^Aa^	7.14 ± 0.19 ^Aa^	6.65 ± 0.22 ^Aa^	6.79 ± 0.15 ^Aa^
	35150	5.66 ± 0.33 ^Ba^	6.09 ± 0.18 ^Ba^	5.91 ± 0.21 ^ABa^	5.84 ± 0.38 ^Ba^
	43889	5.62 ± 0.34 ^Ba^	6.53 ± 0.10 ^ABb^	6.16 ± 0.06 ^ABab^	6.28 ± 0.06 ^Bb^
	43890	5.81 ± 0.07 ^Ba^	6.29 ± 0.28 ^Ba^	5.97 ± 0.50 ^ABa^	5.98 ± 0.51 ^Ba^
	43895	5.77 ± 0.25 ^Ba^	6.19 ± 0.34 ^Ba^	5.83 ± 0.40 ^Ba^	5.67 ± 0.16 ^Ba^
**15 °C**	iso	4.94 ± 0.07 ^Aa^	6.33 ± 0.21 ^Ab^	7.06 ± 0.13 ^ABc^	6.86 ± 0.12 ^Abc^
	35150	3.90 ± 0.42 ^Ba^	5.90 ± 0.32 ^Ab^	6.26 ± 0.17 ^Bb^	6.08 ± 0.18 ^Ab^
	43889	3.75 ± 0.06 ^Ba^	5.10 ± 0.27 ^Bb^	6.29 ± 0.13 ^Bc^	6.21 ± 0.33 ^Ac^
	43890	3.41 ± 0.25 ^Ba^	5.04 ± 0.17 ^Bb^	6.16 ± 0.08 ^Bc^	6.04 ± 0.24 ^Ac^
	43895	4.13 ± 0.25 ^Ca^	5.68 ± 0.69 ^ABb^	6.30 ± 0.38 ^Bb^	6.24 ± 0.34 ^Ab^

Mean ± standard deviation (SD). ^1^ Different uppercase letters in a column represent a significant difference (*p* < 0.05). ^2^ Different lowercase letters in a row represent a significant difference (*p* < 0.05).

**Table 2 foods-14-03091-t002:** Population (log CFU/cm^2^) of biofilm from *E. coli* isolated from the stent and *E. coli* O157:H7 (ATCC 35150, ATCC 43889, ATCC 43890, and ATCC 43895) subjected to 0–180 s UVC treatments ^1,2^.

UVC Treatment Time (s)	Biofilm Maturation Temperature	*Escherichia coli* Strains				
		iso	35150	43889	43890	43895
**0**	37 °C	5.58 ± 0.43 ^Aa^	5.71 ± 0.10 ^Aa^	5.03 ± 0.47 ^Aa^	5.21 ± 0.54 ^Aa^	5.41 ± 0.30 ^Aa^
	25 °C	5.90 ± 0.03 ^Aa^	5.87 ± 0.19 ^Aa^	5.79 ± 0.30 ^Ba^	5.78 ± 0.31 ^ABa^	5.93 ± 0.28 ^Aa^
	15 °C	6.06 ± 0.26 ^Aa^	5.65 ± 0.31 ^Aa^	5.79 ± 0.09 ^Ba^	5.90 ± 0.07 ^Ba^	5.83 ± 0.13 ^Aa^
**30**	37 °C	4.38 ± 0.17 ^Aa^	4.68 ± 0.62 ^Aa^	4.31 ± 0.61 ^Aa^	3.93 ± 0.41 ^Aa^	4.12 ± 0.39 ^ABa^
	25 °C	4.34 ± 0.23 ^Aa^	3.90 ± 0.49 ^Ba^	4.22 ± 0.12 ^Aa^	3.83 ± 0.20 ^Aa^	4.47 ± 0.16 ^Ab^
	15 °C	5.56 ± 0.22 ^Ba^	4.68 ± 0.34 ^Ab^	4.01 ± 0.09 ^Ac^	3.56 ± 0.03 ^Ac^	3.80 ± 0.09 ^Bc^
**60**	37 °C	4.32 ± 0.28 ^Aa^	4.22 ± 0.44 ^Aa^	3.93 ± 0.40 ^Aa^	3.73 ± 0.51 ^Aa^	3.82 ± 0.33 ^Aa^
	25 °C	4.25 ± 0.23 ^Aa^	3.35 ± 0.24 ^Bb^	4.19 ± 0.29 ^Aa^	3.89 ± 0.42 ^Aab^	4.16 ± 0.18 ^Aa^
	15 °C	5.28 ± 0.23 ^Ba^	3.85 ± 0.29 ^ABb^	3.79 ± 0.11 ^Ab^	3.72 ± 0.43 ^Ab^	3.78 ± 0.16 ^Ab^
**90**	37 °C	3.88 ± 0.42 ^Aa^	3.88 ± 0.56 ^Aa^	3.66 ± 0.31 ^ABa^	3.78 ± 0.45 ^Aa^	3.83 ± 0.05 ^Aa^
	25 °C	4.24 ± 0.26 ^ABa^	3.69 ± 0.24 ^Aa^	4.06 ± 0.15 ^Aa^	3.80 ± 0.24 ^Aa^	4.06 ± 0.22 ^Aa^
	15 °C	4.72 ± 0.46 ^Ba^	3.32 ± 0.40 ^Ab^	3.43 ± 0.15 ^Bb^	3.78 ± 0.38 ^Ab^	3.20 ± 0.46 ^Bb^
**120**	37 °C	3.77 ± 0.67 ^Aa^	3.78 ± 0.12 ^Aa^	3.26 ± 0.14 ^Aa^	3.20 ± 0.22 ^Aa^	3.44 ± 0.38 ^ABa^
	25 °C	3.49 ± 0.43 ^Aa^	3.74 ± 0.10 ^Aab^	4.06 ± 0.25 ^Bb^	3.34 ± 0.05 ^ABa^	3.73 ± 0.24 ^Aab^
	15 °C	4.54 ± 0.09 ^Ba^	3.48 ± 0.13 ^Ab^	3.50 ± 0.10 ^ABb^	3.83 ± 0.21 ^Bb^	3.19 ± 0.29 ^Bc^
**150**	37 °C	3.76 ± 0.23 ^Aa^	3.08 ± 0.19 ^Aa^	3.39 ± 0.22 ^Aa^	3.49 ± 0.18 ^Aa^	3.57 ± 0.41 ^Aa^
	25 °C	3.89 ± 0.10 ^ABa^	3.13 ± 0.19 ^Ab^	3.19 ± 0.18 ^Ab^	3.20 ± 0.14 ^Ab^	4.08 ± 0.24 ^Aa^
	15 °C	4.39 ± 0.22 ^Ba^	3.23 ± 0.13 ^Ab^	3.15 ± 0.04 ^Ab^	3.45 ± 0.15 ^Ab^	2.78 ± 0.29 ^Bc^
**180**	37 °C	2.12 ± 0.16 ^Aa^	2.25 ± 0.30 ^Aa^	2.51 ± 0.59 ^Aa^	2.38 ± 0.23 ^Aa^	2.72 ± 0.13 ^Aa^
	25 °C	3.67 ± 0.06 ^Ba^	2.93 ± 0.20 ^Ba^	3.41 ± 0.39 ^Ba^	3.04 ± 0.09 ^Ba^	3.56 ± 0.08 ^Ba^
	15 °C	4.22 ± 0.29 ^Ba^	3.31 ± 0.20 ^Bb^	3.05 ± 0.17 ^ABb^	2.92 ± 0.46 ^ABb^	2.94 ± 0.33 ^Ab^

Mean ± standard deviation (SD). ^1^ Different uppercase letters in a column represent a significant difference (*p* < 0.05). ^2^ Different lowercase letters in a row represent a significant difference (*p* < 0.05).

**Table 3 foods-14-03091-t003:** Quantification of extracellular polysaccharides in the biofilms of *E. coli* isolated from a biliary stent and *E. coli* O157:H7 (ATCC 35150, ATCC 43889, ATCC 43890, and ATCC 43895) matured at 15 °C, 25 °C, and 37 °C ^1,2^.

Biofilm Maturation Temperature	Strain	Biofilm Carbohydrate Content in EPSs (μg/cm^2^)			
		1 Day	3 Day	5 Day	7 Day
**37 °C**	iso	1.33 ± 0.09 ^Aa^	1.64 ± 0.09 ^Ab^	1.53 ± 0.09 ^Abc^	1.46 ± 0.08 ^Ac^
	35150	1.14 ± 0.10 ^Ba^	1.45 ± 0.09 ^BCb^	1.46 ± 0.04 ^ABb^	1.41 ± 0.05 ^Ab^
	43889	1.19 ± 0.12 ^Aba^	1.49 ± 0.13 ^Bb^	1.50 ± 0.02 ^Ab^	1.45 ± 0.07 ^Ab^
	43890	1.17 ± 0.12 ^Ba^	1.34 ± 0.07 ^Cb^	1.40 ± 0.05 ^Bb^	1.43 ± 0.04 ^Ab^
	43895	1.16 ± 0.09 ^Ba^	1.43 ± 0.09 ^BCb^	1.38 ± 0.03 ^Bb^	1.42 ± 0.07 ^Ab^
**25 °C**	iso	1.72 ± 0.03 ^Aa^	1.74 ± 0.03 ^Aa^	1.73 ± 0.03 ^Aa^	1.73 ± 0.02 ^Aa^
	35150	1.53 ± 0.02 ^Ba^	1.54 ± 0.03 ^Ba^	1.56 ± 0.02 ^Ba^	1.57 ± 0.03 ^Ba^
	43889	1.55 ± 0.02 ^Ba^	1.58 ± 0.03 ^Ba^	1.57 ± 0.01 ^Ba^	1.58 ± 0.01 ^Ba^
	43890	1.49 ± 0.02 ^Ba^	1.51 ± 0.02 ^Ba^	1.52 ± 0.02 ^Ba^	1.52 ± 0.03 ^Ba^
	43895	1.55 ± 0.04 ^Ba^	1.54 ± 0.02 ^Ba^	1.57 ± 0.03 ^Ba^	1.55 ± 0.03 ^Ba^
**15 °C**	iso	2.34 ± 0.02 ^Aa^	2.52 ± 0.06 ^Ab^	2.58 ± 0.03 ^Ab^	2.60 ± 0.03 ^Ab^
	35150	1.66 ± 0.02 ^Ba^	1.69 ± 0.02 ^Ba^	1.68 ± 0.01 ^BCa^	1.69 ± 0.01 ^BCa^
	43889	1.70 ± 0.02 ^Ba^	1.72 ± 0.03 ^Ba^	1.72 ± 0.02 ^BCa^	1.70 ± 0.02 ^BCa^
	43890	1.65 ± 0.03 ^Ba^	1.64 ± 0.03 ^Ba^	1.64 ± 0.01 ^Ba^	1.63 ± 0.01 ^Ba^
	43895	1.74 ± 0.02 ^Ba^	1.76 ± 0.02 ^Ba^	1.74 ± 0.02 ^Ca^	1.73 ± 0.02 ^Ca^

Mean ± standard deviation (SD). ^1^ Different uppercase letters in a column represent a significant difference (*p* < 0.05). ^2^ Different lowercase letters in a row represent a significant difference (*p* < 0.05).

## Data Availability

The original contributions presented in this study are included in the article/Appendix A. Further inquiries can be directed to the corresponding author.

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
