# Peer review of "Effects of UVC Treatment on Biofilms of Escherichia coli Strains Formed at Different Temperatures and Maturation Periods"

_foods, 2025, doi:10.3390/foods14173091_

Round 1

Reviewer 1 Report

Comments and Suggestions for Authors

Abstract: Replace vague phrases such as “revealing that peak biofilm formation occurred...” with more direct and precise statements

Some sentences are overly complex or repetitive. Simplifying phrasing and avoiding redundancy (like restating findings already mentioned) can improve readability. This comment applies to the entire manuscript.

Introduction:

Lines 49, 53, 55: Include references

Line 65: replace word “develop” with “grow”

Lines 73-84: relevant to objectives of this statement to highlight problem statements. However, the follow-up paragraphs and prior paragraphs and the transitions make this introduction unfocused.

Line 97: Abbreviate API upon first usage.

Overall: Introduction provided with a general background of E. coli and its biofilm formation aspects. However, Introduction missing the background related to research problem in question. Authors are strongly encouraged to revise the introduction to provide past work and knowledge gaps to pursue current study.

Methodology:

Section 2.1.1: provide details of samples such as number of samples collected, how the microorganisms are dislodged from surfaces, type of test kit used for Illumina sequencing. This entire section requires detailed description of methodology.

Line 146: It is expected several gram-negatives will be grown on MacConkey. How do the authors select a representative colony assuming that could be E. coli in this case?

Section 2.2: API test method procedure requires better description. Many abbreviated terms and their purpose in the test procedure was unclear.

Line 178: Provide details of ratio of bacteria to media. Does PBS is appropriate to use at this stage? Provide rationale.

Line 179: Why incubation at 4C?

Line 201: incomplete

Line 205: reason for choosing this intensity?

Methodology for quantification after UVC treatment is not provided

Overall: Methodology requires significant improvements.

Line 253-254: this provides answers to a question raised earlier. Authors should have explained this in methodology.

Figure 2: Include statistical differences across the bars in the graph

Lines 280-282: How are these conditions of food processing facilities relevant to clinical settings? Since the primary objective of this study was to determine biofilm formation characteristics of clinically relevant organisms

Line 299-300: If this is the case, is your conclusion valid?

Overall, a thorough revision of the manuscript is necessary to create a more focused and well-organized study. This should include detailed descriptions of the procedures, clear rationale for selecting the specific conditions, and an explanation of their relevance and potential applications.

Author Response

Abstract: Replace vague phrases such as “revealing that peak biofilm formation occurred...” with more direct and precise statements

Some sentences are overly complex or repetitive. Simplifying phrasing and avoiding redundancy (like restating findings already mentioned) can improve readability. This comment applies to the entire manuscript.

  Thank you for your comment. The manuscript was edited by the English editing service of MDPI. We also reviewed the manuscript one more time and revised the sentences. Please check the revised manuscript once more.

Introduction:

Lines 49, 53, 55: Include references

Thank you for your comment. Following references were added in the sentences (L 50-54).

  1. Ministry of Food and Drug Safety (MFDS). Foodborne disease outbreak (2025). Available from: https://www.foodsafetykorea.go.kr. Accessed June. 23, 2025.
  2. Shah, H. J. (2024). Reported incidence of infections caused by pathogens transmitted commonly through food: impact of increased use of culture-independent diagnostic tests—foodborne diseases active surveillance network, 1996–2023. MMWR. Morbidity and Mortality Weekly Report73.
  3. U. S. Centers for disease control and prevention (US CDC). E. coli outbreak linked to onions served at McDonald’s (2025). Available from: https://www.cdc.gov/ecoli/outbreaks/e-coli-O157.html

Line 65: replace word “develop” with “grow”

  Thank you for your comment. The word was revised as you recommended (L 67)

Lines 73-84: relevant to objectives of this statement to highlight problem statements. However, the follow-up paragraphs and prior paragraphs and the transitions make this introduction unfocused.

  Thank you for your comment. The sentences were moved, and a new sentence was added to indicate the objective of this study more clearly (L 100-113).

Line 97: Abbreviate API upon first usage.

  Thank you for your comment. Full name of API was indicated as you recommended (L 117)

Overall: Introduction provided with a general background of E. coli and its biofilm formation aspects. However, Introduction missing the background related to research problem in question. Authors are strongly encouraged to revise the introduction to provide past work and knowledge gaps to pursue current study.

  Thank you for your valuable comment. The introduction was significantly revised with recent findings related to this topic. The objective of this study was also represented more clearly as you suggested (L 33-131)

Methodology:

Section 2.1.1: provide details of samples such as number of samples collected, how the microorganisms are dislodged from surfaces, type of test kit used for Illumina sequencing. This entire section requires detailed description of methodology.

Thank you for your comment. The following sentence were added in this manuscript (L 135-149)

Bile samples and biliary stents from patients were collected by Dankook University Hospital, Cheonan, Republic of Korea with ethical approval obtained from the Faculty of Dankook University Hospital Review Board (IRB No: 2023-06-004).

The outside of the stent was washed with 70% ethanol. The washed stent was cut into uniform pieces (approximately 0.5 cm) using a medical scalpel, and vortexed with glass beads.

Among the 4 samples, a sample with the highest quality was further analyzed for the 16S rRNA metagenome sequencing.

Line 146: It is expected several gram-negatives will be grown on MacConkey. How do the authors select a representative colony assuming that could be E. coli in this case?

From the colonies observed on MacConkey agar, three pink colonies and three colorless colonies were selected and identified using the API 20E kit, which is commonly employed for the identification of enteric and other Gram-negative rods. The information was added in the manuscript (L 178-181)

Section 2.2: API test method procedure requires better description. Many abbreviated terms and their purpose in the test procedure was unclear.

  Thank you for your comment. The API test was conducted according to the manufacturer’s instructions. Therefore, details were deleted, and references were added (L185-194).

Line 178: Provide details of ratio of bacteria to media. Does PBS is appropriate to use at this stage? Provide rationale.

PBS has been widely used for the biofilm formation assay. For rationality, recent publications were added in the manuscript (L 215).

Kim, S. H., Jyung, S., & Kang, D. H. (2022). Comparative study of Salmonella Typhimurium biofilms and their resistance depending on cellulose secretion and maturation temperatures. LWT154, 112700.

Line 179: Why incubation at 4C?

This process is widely used for cell attachment during biofilm formation. Reference was added for the rationality (L 217).

Lee, J. I., Kim, S. S., & Kang, D. H. (2023). Characteristics of Staphylococcus aureus biofilm matured in tryptic soy broth, low-fat milk, or whole milk samples along with inactivation by 405 nm light combined with folic acid. Food Microbiology116, 104350.

Line 201: incomplete

  Sorry for the confusing sentence, “wavelength of 253.7 nm was used” in this study (L 239).

Line 205: reason for choosing this intensity?

  Thank you for your comment, but the intensity of this UV lamp was fixed. We measured the irradiation intensity of UVC at 17 cm distance (L 240-241).

Methodology for quantification after UVC treatment is not provided

  Thank you for your comment. For enumeration, cell suspensions were ten-fold serially diluted in 9 ml of PW. After that, 100 µl aliquots were spread-plated onto Sorbitol MacConkey agar (Difco), followed by in-cubation for 24 h at 37 °C. After incubation, typical colonies (pink colonies for isolated E. coli and white colonies for E. coli O157:H7) were counted. This information was added in the manuscript (L 249-251)

Overall: Methodology requires significant improvements.

  Thank you for your kind advice. We revised the materials and methods section significantly. Please check the revised manuscript once more.

Line 253-254: this provides answers to a question raised earlier. Authors should have explained this in methodology.

  Thank you for your kindness. We added additional information to the M&M section for better understanding

Figure 2: Include statistical differences across the bars in the graph

Statistical differences were added in the revised figure 2 as you recommended (L382)

Lines 280-282: How are these conditions of food processing facilities relevant to clinical settings? Since the primary objective of this study was to determine biofilm formation characteristics of clinically relevant organisms

  Thank you for your comment. Even though isolated E. coli strain was from clinical samples, the others are foodborne pathogens. Therefore, we simulated the food processing facilities. Related sentences were added in the manuscript.

Line 299-300: If this is the case, is your conclusion valid?

  Biofilms matured at 15°C exhibited the highest resistance to UVC, which was associated with the greatest exopolysaccharide production.

Overall, a thorough revision of the manuscript is necessary to create a more focused and well-organized study. This should include detailed descriptions of the procedures, clear rationale for selecting the specific conditions, and an explanation of their relevance and potential applications.

  Thank you for your meticulous review comments. English was revised with the editing service and the manuscript was revised significantly considering the reviewers comments. Please check the revised manuscript once more.

Reviewer 2 Report

Comments and Suggestions for Authors

The manuscript submitted by Kyoung & Kim aimed to evaluate the effectiveness of UVC application on biofilm formation by Escherichia coli, considering different temperatures and biofilm maturation times.

Following a critical review of the manuscript, my main question is: what is the rationale behind testing the effect of UVC on a biofilm-forming E. coli strain isolated from bile? Does this application have any practical relevance for the treatment of patients with biliary obstructions? The authors must make this very clear in the discussion section of the manuscript.

Remarks

  • Remove the running title, as it is not required according to the journal's guidelines.
  • Line 12: Specify which type of resistance was evaluated.
  • Line 12: Do not abbreviate Escherichia coli upon its first mention in the abstract.
  • In the keywords section, replace words that already appear in the title with others that better contextualize the study area to improve indexing.
  • Lines 48–53: Include references for the reported number of cases. What is the source of this information?
  • Lines 73–84: The purpose of the in-depth description of bile functions is unclear. The authors need to connect this information to E. coli. Although the strain was isolated from bile, I suggest that the authors include information that links bile functions to the bacterium. As it stands, this information appears disconnected from the rest of the introduction.
  • I suggest that the authors clearly highlight the novelties of this study in the introduction.
  • At the beginning of the Materials and Methods section, the authors should include a flowchart or figure illustrating the experimental design to facilitate understanding. In its current form, this section does not fully convey how the experiment was structured and conducted. If the authors prefer not to use a figure, they should include a dedicated subsection titled "Experimental Design" to describe this information in text form. However, I believe a figure would significantly aid reader comprehension.
  • Lines 122/123: Adjust formatting according to the journal’s style.
  • Line 160 and elsewhere: Standardize the use of the genus name, spelling out Escherichia coli at the first mention in each section, and using the abbreviated form (E. coli) thereafter.
  • Include appropriate references for sections 2.3, 2.5, and 2.6.
  • Line 183: Standardize the use of italics for scientific names throughout the manuscript.
  • After reading the Materials and Methods section, it remains unclear how the number of cells present in the biofilms was quantified, even though this information is reported in the abstract. This must be clarified.
  • Lines 224–227: Add appropriate references.
  • Line 234: Separate the two genera mentioned.
  • Line 238 and elsewhere: Cite the authors in the text in addition to providing the numerical references.
  • Line 244: These are not species; the authors are referring to bacterial genera.
  • Line 264: Provide references for the previous studies mentioned.
  • Table 1 is entirely unnecessary and should be removed.
  • In section 3.2, the authors excessively repeat the results already presented in the table and should reconsider how this information is conveyed in the manuscript.
  • For Table 3, clarify whether the reported values represent population counts or log reductions.
  • The conclusions merely repeat the results. The authors should be more objective in stating the actual conclusions that can be drawn from the study.

Author Response

The manuscript submitted by Kyoung & Kim aimed to evaluate the effectiveness of UVC application on biofilm formation by Escherichia coli, considering different temperatures and biofilm maturation times.

Following a critical review of the manuscript, my main question is: what is the rationale behind testing the effect of UVC on a biofilm-forming E. coli strain isolated from bile? Does this application have any practical relevance for the treatment of patients with biliary obstructions? The authors must make this very clear in the discussion section of the manuscript.

  Thank you for your effort to review our manuscript. We aim to compare the characteristics of biofilms isolated from the clinical sample with the pathogenic E. coli in this study. Our findings suggest that the properties of E. coli isolated from the clinical sample was significantly different from the pathogenic E. coli in laboratory. We revised our manuscript significantly reflecting the reviewers’ comments. Also, the English of manuscript was revised using the English editing service. Please check the revised manuscript once more.

Remarks

  • Remove the running title, as it is not required according to the journal's guidelines.

Sorry for our mistake. We deleted the running title (L 4).

  • Line 12: Specify which type of resistance was evaluated.

Thank you for your valuable comment. In this study, UVC resistance was investigated and indicated in the abstract as you recommended (L 12)

  • Line 12: Do not abbreviate Escherichia coli upon its first mention in the abstract.

Sorry for our mistake. It was indicated the full name at the first mention as you pointed out (L 13)

  • In the keywords section, replace words that already appear in the title with others that better contextualize the study area to improve indexing.

Thank you for your comment. The keywords were revised as you recommended (L 29-30)

  • Lines 48–53: Include references for the reported number of cases. What is the source of this information?

Thank you for your comment. The references were added as you recommended (L 50-54).

  • Lines 73–84: The purpose of the in-depth description of bile functions is unclear. The authors need to connect this information to E. coli. Although the strain was isolated from bile, I suggest that the authors include information that links bile functions to the bacterium. As it stands, this information appears disconnected from the rest of the introduction.

Thank you for your suggestion. We moved this section and added new sentences for the better presentation of the objective of this study (L 100-131).  

  • I suggest that the authors clearly highlight the novelties of this study in the introduction.

  Thank you for your comment. The introduction section was revised significantly to highlight the novelties of this study (L 100-131).

  • At the beginning of the Materials and Methods section, the authors should include a flowchart or figure illustrating the experimental design to facilitate understanding. In its current form, this section does not fully convey how the experiment was structured and conducted. If the authors prefer not to use a figure, they should include a dedicated subsection titled "Experimental Design" to describe this information in text form. However, I believe a figure would significantly aid reader comprehension.

Thank you for your valuable comment. The schematic diagram of experimental design was added as you recommended (L264)

  • Lines 122/123: Adjust formatting according to the journal’s style.

Thank you for your comment, but this style was edited by journal (Foods). We revised the format once more (L 154-155)

  • Line 160 and elsewhere: Standardize the use of the genus name, spelling out Escherichia coli at the first mention in each section, and using the abbreviated form (E. coli) thereafter.

Thank you for your suggestion, but we thought the first introduction of full name in the abstract and introduction section would be okay.

  • Include appropriate references for sections 2.3, 2.5, and 2.6.

Thank you for your comment. We added references for the sections (2.3, 2.5, and 2.6)

  • Line 183: Standardize the use of italics for scientific names throughout the manuscript.

Sorry for our mistake. The name of microorganisms was italicized as you pointed out.

  • After reading the Materials and Methods section, it remains unclear how the number of cells present in the biofilms was quantified, even though this information is reported in the abstract. This must be clarified.

Thank you for your comment. The enumeration methods were added as you pointed out (L 248-251)

For enumeration, cell suspensions were ten-fold serially diluted in 9 ml of PW. After that, 100 µl aliquots were spread-plated onto Sorbitol MacConkey agar (Difco), followed by in-cubation for 24 h at 37 °C. After incubation, typical colonies (pink colonies for isolated E. coli and white colonies for E. coli O157:H7) were counted.

  • Lines 224–227: Add appropriate references.

Thank you for your comment. We added references for this sentences.

  • Line 234: Separate the two genera mentioned.

Thank you for your comment. The two genera were separated as you recommended (L 279)

  • Line 238 and elsewhere: Cite the authors in the text in addition to providing the numerical references.

Thank you for your comment. We added the name of authors as you recommended (L 284 et al.)

  • Line 244: These are not species; the authors are referring to bacterial genera.

Thank you for your valuable comment. The word was revised as you pointed out (L 290)

  • Line 264: Provide references for the previous studies mentioned.

Sorry for the confusing sentence. We aimed to describe that the subsequent experiments (the other experiments in this study) were conducted. We revised the sentence correctly (L 311)  

  • Table 1 is entirely unnecessary and should be removed.\

Thank you for your comment. We uploaded Table 1 as supplementary file for the readers who needed it (L 317).

  • In section 3.2, the authors excessively repeat the results already presented in the table and should reconsider how this information is conveyed in the manuscript.

Thank you for your comment, but we decided that the description in the manuscript is needed in section 3.2.

  • For Table 3, clarify whether the reported values represent population counts or log reductions.

We indicated that the populations (log CFU/cm2) of E. coli strains were represented in this table (Table 3)

  • The conclusions merely repeat the results. The authors should be more objective in stating the actual conclusions that can be drawn from the study.

Thank you for your comment. The conclusion section was revised significantly as you recommended (L 533-544).

Round 2

Reviewer 1 Report

Comments and Suggestions for Authors

some figures in the revised version are still missing significant figures. Please check. 

Author Response

Comments: some figures in the revised version are still missing significant figures. Please check. 

Answer: Thank you for your comment. The manuscript was revised correctly. 

Reviewer 2 Report

Comments and Suggestions for Authors

All my corrections were fully adressed and the manuscript has improved.

Author Response

Q) All my corrections were fully adressed and the manuscript has improved.

A) We revised the manuscript once more. Please check the final manuscript.